# Controllable Procedural Generation of Landscapes

Jia-Hong Liu
Tsinghua University
Beijing, China
liujiaho23@mails.tsinghua.edu.cn

Chuyue Zhang
Tsinghua University
Beijing, China
zhangchu23@mails.tsinghua.edu.cn

Shao-Kui Zhang[*]
Tsinghua University
Beijing, China
shaokui@tsinghua.edu.cn

Song-Hai Zhang
Key Laboratory of Pervasive Computing
Ministry of Education
Beijing, China
Tsinghua University
Beijing, China
shz@tsinghua.edu.cn

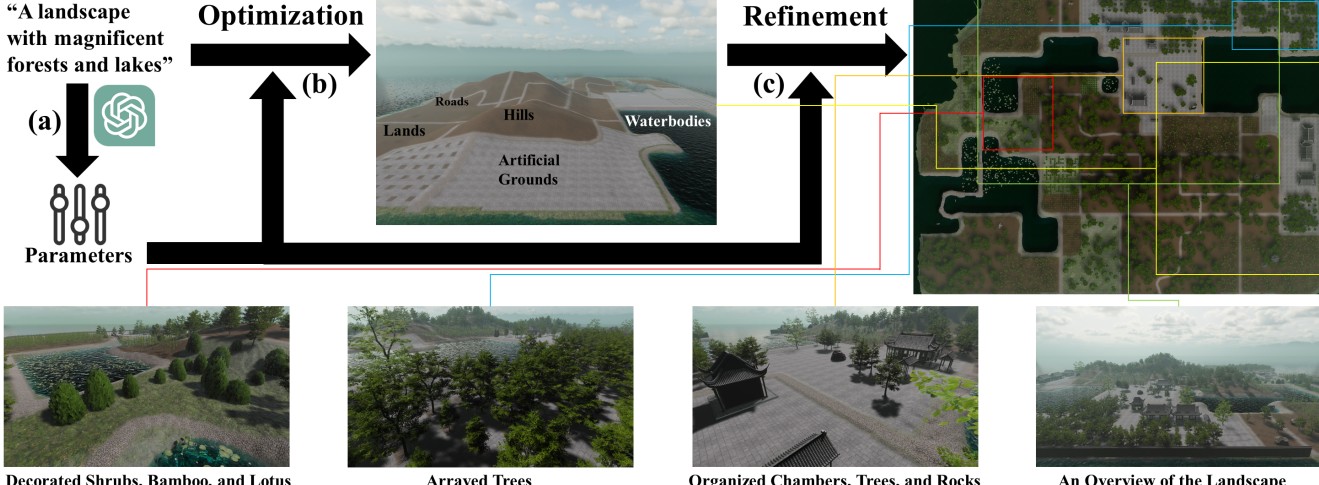

**Figure 1: We propose a controllable and practical framework that generates landscapes. (a) Our framework integrates a Large Language Model that converts user input into parameters and enables user control over the generation process through simple texts. (b) It leverages optimization techniques to generate a general plan that includes terrains, roads, and various attributes. (c) Subsequently, it refines the plan by smoothing and arranging elements through rules and patterns for aesthetic landscapes.**

## Abstract

Landscapes, recognized for their indispensable role in large-scale scenes, are experiencing growing demand. However, the manual modeling of such content is labor-intensive and lacks efficiency. Procedural Content Generation (PCG) techniques enable the rapid generation of diverse landscape elements. Nevertheless, ordinary users may encounter difficulties controlling these methods for desired results. In this paper, we introduce a controllable framework for procedurally generating landscapes. We integrate state-of-the-art Large Language Models (LLMs) to enhance user accessibility and control. By converting plain text inputs into parameters through LLMs, our framework allows ordinary users to generate a batch of plausible landscapes tailored to their specifications. A parameter-controlled PCG procedure is designed to leverage optimization techniques and employ rule-based refinements. It achieves harmonious layering in terrains, zoning, and roads while enabling aesthetic arrangement of vegetation and artificial elements. Extensive experiments demonstrate our framework's effectiveness in generating landscapes comparable to those crafted by experienced architects. Our framework has the potential to enhance the productivity of landscape designers significantly[1].

[*]Corresponding Author

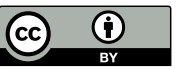

MM '24, October 28-November 1, 2024, Melbourne, VIC, Australia
© 2024 Copyright held by the owner/author(s).
ACM ISBN 979-8-4007-0686-8/24/10
https://doi.org/10.1145/3664647.3681129

[1]The source code is publicly available at:
https://github.com/omegafantasy/ControllableLandscape, and
https://github.com/omegafantasy/ControllableLandscape_UnityClient.

## CCS Concepts

• **Computing methodologies** → **Computer graphics**; *Modeling and simulation*; *Search methodologies.*

## Keywords

Landscape Planning, Procedural Generation, Large Language Model

**ACM Reference Format:**
Jia-Hong Liu, Shao-Kui Zhang, Chuyue Zhang, and Song-Hai Zhang. 2024. Controllable Procedural Generation of Landscapes. In *Proceedings of the 32nd ACM International Conference on Multimedia (MM '24), October 28-November 1, 2024, Melbourne, VIC, Australia.* ACM, New York, NY, USA, 10 pages. https://doi.org/10.1145/3664647.3681129

## 1 Introduction

Recently, there is a growing interest in developing efficient approaches for generating 3D indoor [62, 63, 65] and outdoor [32, 43, 48] content. Landscapes are indispensable outdoor content that creates ecologically and aesthetically balanced living spaces, promoting healthy lifestyles in modern life [37, 64]. However, architects often rely on modeling software to craft landscapes, which lacks efficiency. Even with powerful tools [26], the modeling process still requires many manual operations and much proficiency [18].

In the industry, generating landscapes is often assisted with Procedural Content Generation (PCG) techniques. PCG facilitates the efficient and large-scale creation of natural environment contents using mathematical rules, spatial patterns, or other techniques such as optimization and deep learning [6, 14, 35, 50]. Nevertheless, these approaches may have limitations in user control. Ordinary users may find it challenging to leverage the process to achieve their objectives fully. When confronted with multiple parameters, users may experience difficulty adjusting them for optimal results. Additionally, some PCG approaches require expertise in engineering or architectural techniques and compromise accessibility to the general audience. Addressing user control is essential for achieving more effective landscape scene generation.

Large Language Models (LLMs) excel in natural language understanding and task performance [1, 52, 53]. Additional context or prompts enable more appropriate and accurate responses aligned with user specifications [52]. Therefore, LLMs are effective and friendly tools for ordinary users to control expert tasks.

In this paper, we propose a controllable framework for procedurally generating landscapes, aiming to generate aesthetic landscapes effectively (Figure 1). We address user control by introducing a set of parameters and integrating an LLM into our framework. Utilizing our framework is simple for ordinary users, who merely need to describe their desired landscapes in plain text. The framework automatically engages the LLM to comprehend the input and provide responses that can be parsed into parameters, thereby guiding the generation process to align with user objectives.

After determining the parameters, we generate the landscape plan leveraging optimization techniques, featuring an adapted genetic algorithm for terrain and vegetation types and a heuristic algorithm for road generation. We use a discrete grid to represent terrain, entrances, scenic spots, roads, and the layout scheme of the landscape. Subsequently, we refine the plan through mathematical smoothing for more natural transitions both horizontally and vertically. Finally, we arrange diverse landscape elements to complete an aesthetic landscape. Furthermore, the framework supports the batch generation of many diverse landscapes given a user input.

We evaluate our framework through experiments. Performance analysis and ablation studies delve into several components and validate the overall plausibility of our method. The user study demonstrates that the landscapes generated by our framework are comparable to those crafted by experienced architects.

The key contributions of this paper are outlined below:

- We propose a framework facilitating the procedural generation of a large number of plausible landscapes.
- Our framework converts text inputs to parameters using a Large Language Model, enhancing the generation process's controllability and accessibility to ordinary users.
- Our framework follows a parameter-controlled procedure combining optimization and rule-based techniques, such as a 2D-adapted genetic algorithm, a heuristic road generation algorithm, and mathematical smoothing.

## 2 Related Works

### 2.1 Procedural Generation of Outdoor Contents

Procedural Content Generation (PCG) is a leading technique owing to its versatile applications across various industries (e.g., gaming) [4, 17, 20, 45]. It is particularly effective for generating modular, highly repetitive, or rule-based content [14]. Traditional methods often rely on rules and simple randomized algorithms [14], such as grammar trees [56] and noise. Recently, more advanced rule-based methods have been proposed, such as binary [29] or multi-level [61] partitioning for urban space planning. Computational geometry [9] and example-based methods [66] are also widely employed. Infinigen [43] is a state-of-the-art approach that facilitates generating natural objects and scenes through randomized mathematical rules.

Other PCG categories are also emerging. Optimization-based methods typically perform multi-objective optimization by integrating the problem into existing algorithmic frameworks. There are cases for integer programming [39], Particle Swarm Optimization [51], anisotropic shortest path algorithm [16], and genetic algorithms [40]. Simulation-based methods simulate temporal changes in content influenced by the autonomous actions of creatures and natural phenomena to generate scenes iteratively. For instance, simulating the interaction of individuals [47, 55] or the natural weathering and battling events [28]. Learning-based methods apply frameworks such as fully connected neural networks [30], Variational Autoencoders (VAEs) [57], and Generative Adversarial Networks (GANs) [19, 23, 34], etc.

Our framework combines the rule-based and optimization-based approaches to leverage their respective advantages. Additionally, incorporating an LLM enhances controllability and accessibility to ordinary users. Recent studies [31, 48] also investigate the integration of LLM in PCG.

### 2.2 Large Language Models

Large language models (LLMs) have garnered extensive attention recently. Many models are introduced [5, 33, 42, 59]. Literature demonstrates their exceptional capabilities. Wei et al. [52] discover

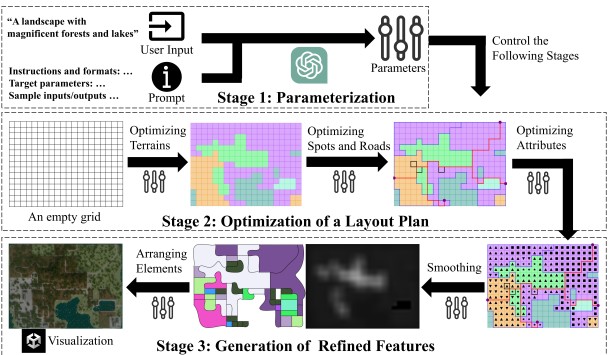

**Figure 2: The framework involves three stages: In the first stage, the LLM is employed to convert the user input into parameters that control the generation. Next, we optimize the landscape's layout plan on a grid, focusing on the terrains, spots, roads, and attributes. Finally, we refine it by smoothing the resulting plan and arranging the elements to obtain an aesthetic and plausible landscape.**

that these models can handle complex tasks even in a zero-shot manner. They [53] also observe that larger scales empower LLMs with additional capabilities compared to smaller models. Bubeck et al. [1] evaluate GPT-4 and conclude that beyond mastering language, it can tackle diverse tasks spanning mathematics, coding, medicine, and more. Typical examples include the advances in reasoning tasks [54] and public health aiding [24]. Numerous applications have emerged based on LLMs, such as Codex [3] and Galactica [49]. This paper employs the GPT-4 [33] model (Section 4.1).

## 2.3 Landscape Generation

Existing literature employs a diverse range of approaches to tackle landscape generation. For instance, PCG techniques are leveraged to automatically model urban forestry [32] and multi-biome landscapes [13]. The genetic algorithm is applied for systematic optimizations [12, 46]. Parametric design approaches [2, 58] are also explored. Simulations are performed to form ecoclimates [36], terrain [7], and glaciers [8]. Real-world data and machine learning [22] are combined to author large-scale ecosystems.

There is growing interest in interactively editing landscapes. For instance, Emilien et al. [11] and Gain et al. [15] facilitate editing virtual worlds using "brush" operations. Schott et al. [44] enable user control over the simulation of large-scale erosion. Similar interactive simulations are also applied to meandering river networks [38]. Ecormier et al. [10] propose a method for interactively generating ecosystems containing flora and fauna. Makowski et al. [27] explore interactive multi-scale modeling of plant ecosystems.

3D-GPT [48] has been proposed as an LLM-powered 3D procedural modeling system that enables the generation of natural scenes through simple instructions. Compared with it, our framework highlights arranging landscape elements rather than modeling them individually. We also support larger quantities of vegetation.

## 3 Overview

Our framework comprises three stages: parameterization, optimization of a layout plan, and generation of refined features (Figure 2). In the first stage (Section 4.1), the LLM converts the user input into parameters. We design the context (prompt) of the queries to ensure accurate and robust responses, which are then parsed into parameters. These parameters cover diverse aspects and functionalities (see Table 1) and guide the subsequent generation.

The second stage (Section 4.2) encompasses three optimization steps, deriving a "layout plan" including information of terrains, spots, roads, and attributes on a grid. The first and third steps employ an adapted genetic algorithm (Section 4.2.2), incorporating modified crossover/mutation operations and a novel "evolution" operation to produce solutions effectively. The first step (Section 4.2.3) assigns each grid cell of five terrain types to characterize its topography and surface features. The second step (Section 4.2.4) first determines entrances and points of interest. Then, a heuristic-based path-finding algorithm and evaluations are applied to generate primary and secondary roads. Like the first step, the third step (Section 4.2.5) assigns each cell one of five attribute types, describing what should be placed on the terrain.

The third stage (Section 4.3) uses two steps to finalize the generation: smoothing (Section 4.3.1) and procedural arrangement of elements (Section 4.3.2). The smoothing step operates in both the horizontal and vertical aspects. Horizontally, it smooths the borders of the regions using cubic splines. Vertically, it generates a smooth height map employing interpolations and Perlin Noise. The second step arranges numerous landscape models based on specified rules and patterns, where factors like model categories, terrain, and location are considered for aesthetic arrangement.

## 4 Method

### 4.1 Parameterization from the Input Text

In this section, we introduce the methodology of utilizing the LLM to convert the user input into parameters. Specifically, we merge the input text with a predefined context, call the LLM API, and parse the response to extract parameters.

We use the GPT-4's chat model [33] $L$, which supports a text-based context comprising one or multiple messages. According to the official guidance, each message is assigned one of three "roles". A "user" message $m_u$ signifies the request from the user, and an "assistant" message $m_a$ represents the response from the model. A "system" message $m_s$, or the system prompt, is an independent message providing instructions and defining the assistant's behavior. Recognizing that a comprehensive context enhances quality and robustness of the responses, we design a context that includes all three message types: a system prompt $m_s$, a sample input $m_u$, and a sample output $m_a$. Section A.1 of the supplementary material provides more details for designing the system prompt. For each query, this context is combined with user input text $m_i$ and sent to the LLM. Therefore, calling the LLM can be expressed as $m_o = L(m_s, m_u, m_a, m_i)$, where $m_o$ is the response text from the model. The output $m_o$ is then parsed into the parameters using a syntax tree, enforcing a dictionary-like format for successful parsing. For $n_p$ parameters $\{p_1, p_2, ..., p_{n_p}\}$, the parsing process is formulated as $\{p_1, p_2, ..., p_{n_p}\} = P(m_o)$.

**Table 1: The list of categorized parameters employed in the paper, with each row representing a category of parameters sharing similar functionalities. Despite some groups appearing similar, they exhibit distinct representations or objectives. For instance, $P_A$ addresses all regions of the same type, whereas $P_a$ targets individual regions.**

| Name | Value Type | Brief Description | Example |
|---|---|---|---|
| $P_\epsilon$ | Boolean | Whether specific elements/types should exist | $P_\epsilon(Highland) = False$ |
| $P_N$ | Pair of Integer | The range (lower/upper limits) of the number of specific regions/elements | $P_N(Lake) = [2, 2]$ |
| $P_A$ | Pair of Float | The range of the total area coverage rates of specific regions | $P_A(Forest) = [0.35, 0.6]$ |
| $P_a$ | Pair of Float | The range of the area coverage rate of any single specific region | $P_a(Lake) = [0.04, 0.2]$ |
| $P_S$ | Pair of Float | The range of the sizes of specific elements | $P_S(RoadWidth) = [2.5, 4]$ |
| $P_E$ | Enumerable | Abstract representation of the extent of specific elements | $P_E(TreeDensity) = Extent.High$ |

Since this paper deals with many parameters, we categorize all parameters into distinct groups (Table 1), where parameters within a category share similar functionalities. Given the complexity of managing all these parameters in a single query, we opt for a multi-agent approach. For every input, we construct multiple queries to the LLM, with each query focusing on parameters related to one specific aspect. Every context $\{m_s, m_u, m_a\}$ used in these queries contains similar background information and adheres to a standardized format. The task description within each context is customized to address the parameters associated with it.

### 4.2 Optimization of a Layout Plan

*4.2.1 High-level Ideas.* With multiple factors (e.g., style, balance, and accessibility) to consider [60] and numerous natural/artificial elements to arrange, generating a landscape in a single-shot process is challenging. Therefore, we address the task in a general-to-specific manner, encompassing two stages. This section (Section 4.2) is the first stage, which focuses on devising a general plan. To reduce the search space, we employ a $w \times h$ grid to formulate the plan discretely.

We adopt a three-step approach that addresses several layers to determine the grid contents. First, we generate the terrain, establishing it as the fundamental feature of the landscape. Second, we determine the layout of key spots (entrances and points of interest) and roads. Lastly, we assign the "attributes", guiding the placement of natural elements or artificial features.

*4.2.2 Genetic Algorithm on a Grid.* The genetic algorithm is a versatile optimization technique commonly employed to address optimization problems. Each potential solution is encoded and treated as an individual. By evaluating these individuals using a fitness function and applying selection, crossover, and mutation operations, the algorithm iteratively fosters new generations of refined solutions to the given problem.

We implement the genetic algorithm on a grid with $w \times h$ cells, where the "types" assigned to all cells represent a solution. Each cell's type is denoted by a non-negative integer, with a total of $K$ types available. Conventional practice employs a 1D binary string encoding for each individual [21]. However, when conforming the 1D encoding to 2D cells, the crossover/mutation operations may lead to fragmented and less optimal solutions due to mismatched spatial properties of 1D strings and 2D grids. Therefore, we introduce specialized modifications to these operations tailored to a grid.

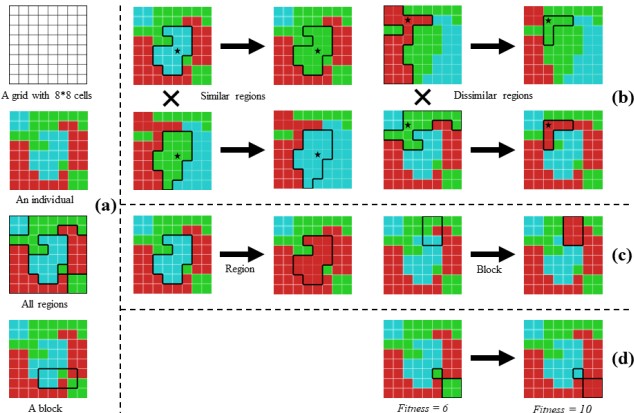

**Figure 3: Illustrations for Section 4.2.2. (a) The basic concepts. (b) Crossover on two cases. For similar selected regions, all contents are exchanged; otherwise, only the intersection parts are exchanged. (c) Mutation on two cases. Contents either in a region or a block are switched as a whole. (d) The novel "evolution" operation. Depending on the fitness function, it actively increases the fitness value.**

Additionally, we propose incorporating an "evolution" operation into the genetic algorithm to accelerate the iterations. These operations are illustrated in Figure 3. Thus, the genetic algorithm follows an iterative selection-crossover-mutation-evolution procedure. Section A.3 of the supplementary material has more details.

*4.2.3 Optimization of the Terrain.* As the first step in Section 4.2, this part determines the terrain for the landscape. Specifically, we assign each cell a terrain type. Five distinct types are considered, each represented by an integer index: unused (0), aquatic (1), terrestrial (2), artificial (3), and elevated (4). These types account for the physical features of the land, encompassing its topography and surface attributes. To elaborate on each type:

- An **"unused"** cell indicates that the cell is excluded from the site, allowing the landscape's shape to vary.
- An **"aquatic"** cell signifies the presence of a water body.
- **"Terrestrial"** and **"artificial"** cells represent flat areas. "Terrestrial" features land with natural elements (soil, rocks, etc.), and "artificial" features ground covered by artificial elements (bricks or cement).

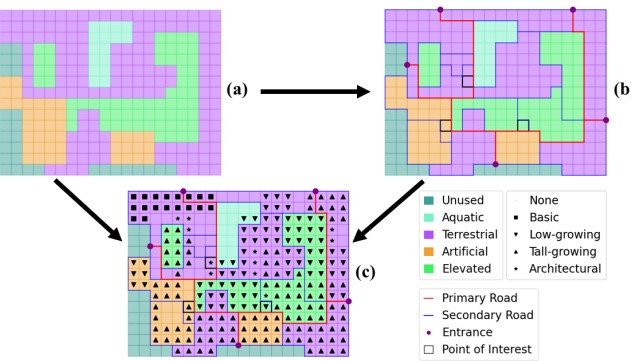

**Figure 4: The results for three steps in Section 4.2. (a) The first step (Section 4.2.3) optimizes terrain types (unused, aquatic, terrestrial, artificial, and elevated) using the genetic algorithm. (b) The second step (Section 4.2.4) determines entrances, points of interest, and two-level roads connecting them. (c) The last step (Section 4.2.5) applies the genetic algorithm again to optimize the attributes (none, basic, low-growing, tall-growing, and architectural).**

- An "**elevated**" cell specifies the presence of a highland or a hill mainly reserved for natural elements.

As previously stated, the genetic algorithm is utilized for the optimization. The fitness function considers multiple factors (existence, numeracy, area, location, and compatibility), detailed in Section A.4 of the supplementary material. Figure 4(a) shows an optimized result in this step.

*4.2.4 Optimization of Spots and Roads.* In the second step, we determine the spots and generate the roads. The generation sequentially targets entrances, points of interest, and primary/secondary roads. A **select-best structure** is applied several times: Randomly or procedurally generate a total of $N_g$ valid solutions, evaluate each of them, and then select the best-evaluated solution.

An entrance is defined as a corner on the boundary of the landscape. In the select-best procedure to generate $P_N(Entrance)$ entrances, solutions with entrances better adhere to the following criteria are more favored: (1) Adequate spacing between pairs, (2) Even distribution across all directions, and (3) Adjacency to flat lands. Similarly, we determine the $P_N(PointOfInterest)$ points of interest (POIs), which serve as scenic spots or landmarks.

Regarding road generation, we prioritize the primary roads, which serve as vital routes connecting key locations. In our context, we generate primary roads that connect all entrances and POIs. We start by connecting every entrance to its nearest POI, then proceed to repeatedly connect pairs of separated POIs with the smallest Euclidean distance until all POIs are connected. We aim for an approach that balances preserving the spatial structure and generating straightforward roads, guided by the following principles [25]: (1) Avoid self-intersections or loops, (2) Follow the region borders when feasible, (3) Minimize unnecessary twists and turns, and (4) Avoid running along the site boundary.

Based on these principles, each road is generated using a randomized heuristic approach, similar to a path-finding process. A

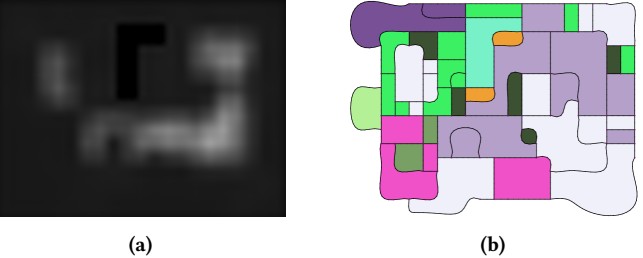

**Figure 5: The results of Section 4.3.1. (a) Vertically, we generate a continuous height map. Lighter areas are higher. (b) Horizontally, we introduce curves to the region borders, with colors signifying various combined types (see Section 4.3.2).**

select-best structure is employed to evaluate and refine the generated roads. After determining all primary roads, we supplement the secondary roads to connect more regions and spots. Section A.5 of the supplementary material provides details for generating these roads. We conclude this part by providing an example of the optimized result, as shown in Figure 4(b).

*4.2.5 Optimization of Attributes.* The plan for vegetation and artificial elements has yet to be decided. In the final step, we address this by assigning each cell an "attribute" type to describe what should be placed on the terrain. Five types are designed: none (0), basic (1), low-growing (2), tall-growing (3), and architectural (4).

- A "**none**" cell indicates that no explicit development is needed for the cell, leaving it as raw terrain.
- A "**basic**" cell recommends fundamental and typical vegetation for the terrain, such as grass, weeds, and rocks.
- "**Low-growing**" and "**tall-growing**" cells feature plants of different heights. Shrubs are typical low-growing vegetation, while trees and bamboo are tall-growing.
- An "**architectural**" cell emphasizes artificial elements such as pavilions, corridors, and statues.

The genetic algorithm is once again utilized. Compared with Section 4.2.3, we consider similar perspectives (existence, numeracy, area, location, and compatibility) but focus more on the attribute's compatibility with the terrain and key spots (Section A.6 of the supplementary material). The result is shown in Figure 4(c).

### 4.3 Generation of Refined Features

*4.3.1 Smoothing.* This section completes the generation by refining the features and determining all elements using two steps: smoothing and procedural arrangement. In the first step, we apply smoothing for a more natural landscape. In the vertical aspect, we generate a continuous height map that specifies the physical height of the ground at any location. The method involves assigning a relative height to each cell based on its terrain type, applying bilinear and bicubic interpolation techniques, and adding Perlin Noise [41]. We then proceed to the horizontal aspect and smooth the region borders by introducing cubic splines. These smoothing techniques are detailed in Section A.7 of the supplementary material. Figure 5 illustrates the results of this step.

**Table 2: The representative elements for the "combined" types. A "/" indicates that the type is invalid or does not require vegetation. "(s)" implies only a few elements, while "(l)" for numerous elements. Note that more elements are supported when arranging the models (in Section 4.3.2). For instance, statues, hallways, and pavilions are all compatible with the type $(T_3, A_4)$.**

|  | Unused ($T_0$) | Aquatic ($T_1$) | Terrestrial ($T_2$) | Artificial ($T_3$) | Elevated ($T_4$) |
|---|---|---|---|---|---|
| None ($A_0$) | / | / | grass (s) | tiles (l) | rocks (s) |
| Basic ($A_1$) | / | lotus (s) | grass (l), shrubs (s) | tiles (l), flowerbeds (s), shrubs (s) | grass (l), rocks (s) |
| Low-growing ($A_2$) | / | lotus (l) | shrubs (l), trees (s) | tiles (l), shrubs (l), flowers (s) | shrubs (l), grass (s), rocks (s) |
| Tall-growing ($A_3$) | / | / | trees (l), bamboos (l) | tiles (l), trees (l) | trees (l), bamboos (l) |
| Architectural ($A_4$) | / | rocks (s) | pavilions (s), trees (l) | tiles (l), chambers (l), trees (s) | pavilions (s), trees (l) |

*4.3.2 Procedural Arrangement of Landscape Elements.* Aiming for a reasonable arrangement, we first determine the "combined" types, referring to the combination of a terrain type and an attribute type. Formally, the combined type of a cell with terrain type $T_i$ and attribute type $A_j$ is denoted as $C_{i,j} = (T_i, A_j)$. It can also be characterized by representative natural and artificial elements, as presented in Table 2. A group of cells sharing the same combined type is defined as a "zone", a refined version of "region" that operates as a cohesive unit with unified functionality and characteristics.

This step finalizes the generation by procedurally arranging landscape models. The arrangement starts with the instantiation of the zone borders, considering different cases for walls, bridges, and roads. Then, we place models within zones as the content, following specific "rules" and "patterns". In our context, a "rule" is represented by the spatial relationship among different models or between models and the zone, referring to spatial relationship explained in [62]. A "pattern" emphasizes particular structures (e.g., arrayed, random, clustering) for placing groups of models, similar to [32]. Rules are typically applied to artificial elements, whereas patterns are primarily used for natural elements. We can combine these rules and patterns for more complicated and appealing landscapes. See Section A.8 of the supplementary material for more details.

The arrangements results are stored as files containing all necessary information (the height map, textures, and model placement) to construct the landscape scene. A Unity plug-in is developed to parse these files and convert them into Unity-native scenes. Each scene is automatically rendered with twenty-one images, comprising an overview from a bird's eye perspective, four large-scale views covering all sides, and sixteen small-scale views from predefined angles and locations. Examples of rendered results and viewpoints are presented in Figure 6. Section B.1 of the supplementary material shows more results and implementation details.

## 5 Experiments

### 5.1 Performance

*5.1.1 Efficiency of the Framework.* The grid size significantly influences the efficiency of our framework. Since the parameterization is LLM-related and unstable, we focus on assessing the optimization (Section 4.2) and refining (Section 4.3) sections. We conduct five hundred trials using widths and lengths from $\{10, 15, 20, 30\}$, with mean execution times in seconds displayed in Figure 7b. Larger grids like $30 \times 30$ require significant time, whereas lower-resolution grids may compromise generation quality. Therefore, we recommend a $20 \times 20$ grid, which takes about twenty seconds with parallel

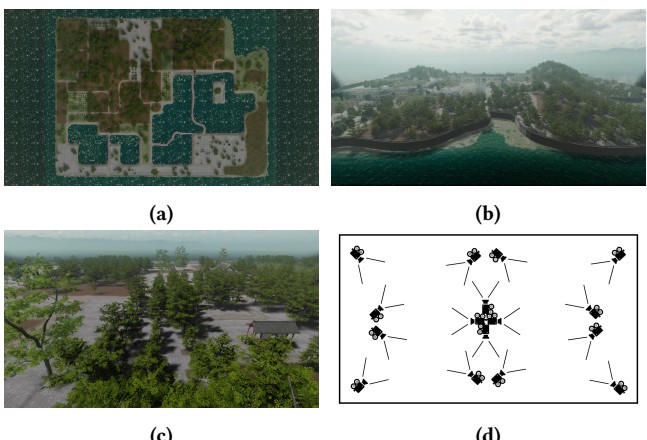

(a)         (b)

(c)         (d)

**Figure 6: We develop a Unity plug-in to automatically convert the results to Unity scenes and render them. (a) A bird's eye view of the whole landscape. (b) A large-scale view from one side of the landscape. (c) A small-scale view within the landscape. (d) Sixteen viewpoints relative to the landscape.**

computing. Using a $20 \times 20$ grid, we analyze the efficiency of various framework components, illustrated in Figure 7a. The components with genetic algorithms and the procedural arrangement consume significant time due to the extensive iterations and computations.

*5.1.2 Influence of the User Input.* In this part, we investigate the influence of user input on the results. First, we assess the LLM's ability to interpret user descriptions. We examine its accuracy in determining the existence of a lake (i.e., $P_\epsilon(Lake)$) based on positive/neutral/negative descriptions (e.g., "The landscape has two lakes", "The site is mostly covered by trees", and "The landscape has no lakes"). Although the LLM may sometimes fail to accurately comprehend when disturbed by additional information, implicit references, or negative wording, the responses are generally accurate. Section B.2 of the supplementary material has more details on this case. For more complicated cases associated with many parameters, we find the LLM able to handle the preferences regarding each parameter accurately. We also test ambiguous or even unrelated inputs (e.g., "An economical landscape"), where parameters are mostly left as default. In general, we conclude that the LLM is effective in input comprehension.

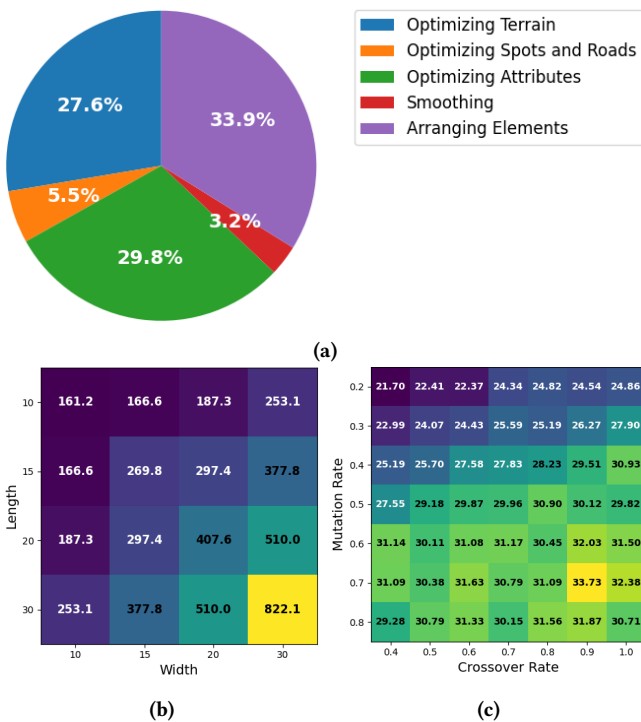

**(a)**

**(b)**  **(c)**

**Figure 7: Some illustrative results from experiments. (a) Time breakdown for framework components, with element arrangement and the genetic algorithm consuming the majority of time. (b) Total execution time across various grid sizes. (c) Maximum fitness values for the genetic algorithm under different crossover and mutation rates. Optimal rates ($\eta_c = 0.9, \eta_m = 0.7$) are chosen as default hyperparameters.**

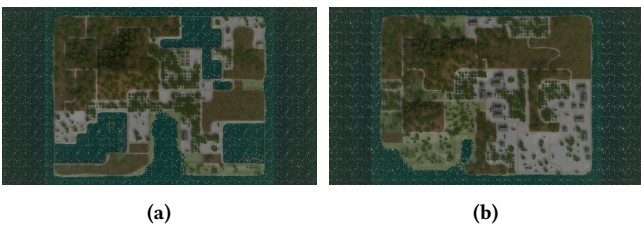

**(a)**  **(b)**

**Figure 8: An example demonstrating the influence of the user input. (a) The user specifies the presence of many lakes. (b) The user specifies that no lakes should exist.**

Then, we examine the generation process's ability to adhere to the parameter. For the case regarding the existence of lakes, we conduct one hundred trials and find that 98% of the resulting landscapes have no lakes when specifying $P_\epsilon(Lake) = False$. Conversely, all landscapes contain at least one lake when $P_\epsilon(Lake)$ is set to True or left unspecified, due to the emphasis on lakes as essential elements in the fitness function. Figure 8 illustrates examples of landscapes with and without lakes. We also test cases for other landscape elements and obtain similar results, which indicates our process's high reliability.

## 5.2 Ablation Study

*5.2.1 Effectiveness of the Genetic Algorithm.* In this part, we evaluate the proposed generic algorithm from several aspects. First, we conduct tests with different crossover rates $\eta_c \in [0.4, 1]$ and mutation rates $\eta_m \in [0.2, 0.8]$. The maximum fitness value for evaluating the terrains (Section 4.2.3) is used as the metric. For each combination of rates, we run the algorithm two hundred times under identical configurations. Figure 7c presents the results. The combination ($\eta_c = 0.9, \eta_m = 0.7$) yields the highest fitness value, indicating that the algorithm is most effective under these values. These values are thereby chosen as the default hyperparameters.

Next, we examine the novel "evolution" operation by comparing our algorithm with the baseline without this operation (i.e., evolution rate $\eta_e = 0$). Using the same experimental settings, the mean fitness value for the baseline yields 26.15, which is 22.5% lower than the value of 33.73 achieved with the evolution operation. This result is anticipated as the evolution operation actively raises fitness values. However, we observe that a high $\eta_e$ may excessively reduce diversity due to extensive modifications. Thus, a default value of $\eta_e = 0.1$ is selected to balance effectiveness and diversity. Finally, we compare the proposed 2D-adapted algorithm with the classic 1D implementation. The classic approach achieves a mean fitness value of 12.74, which is significantly outperformed by our method. It demonstrates that the proposed genetic algorithm successfully adapts to the 2D case and produces better results.

*5.2.2 Effectiveness of the Optimization of Roads.* In this part, we compare the heuristic road generation approach in Section 4.2.4 with a baseline approach that connects corners with at most two straight segments (in an 'L' shape). Both methods are tested on random terrains, entrances, and POIs to generate the primary roads until all entrances and POIs are connected. The results demonstrate that although our method produces slightly longer roads, it significantly reduces the disruption to the region structure and results in more balanced region areas. See Section B.4 of the supplementary material for details.

## 5.3 User Study

*5.3.1 Task Specification.* This section elaborates on the user study that evaluates our framework. We invite three experienced designers to craft landscapes manually within Unity. All landscapes are configured as $400 \times 300$ meters (equivalent to a grid size of $20 \times 15$), with the same set of assets utilized for design. Ten landscape scenes are obtained for comparison (**Manual**), each taking about two days to complete. Furthermore, we design two baseline approaches modified from our framework. The first baseline (**Random1**) replaces the optimization of attributes (Section 4.2.5) with a naive process that assigns each cell a random attribute. The second one (**Random2**) simplifies the procedural arrangement of elements (Section 4.3.2) by organizing all elements in a randomized manner. These baselines also serve as a supplement to the ablation study. One hundred landscapes are randomly generated for each baseline and our method (**Ours**).

Six criteria with assigned weights, devised by professional landscape architects, are employed to facilitate a comprehensive evaluation (Table 3). These criteria are explained in Section B.5 of the

**Table 3: The criteria for comprehensively evaluating the landscapes, weighted based on their importance.**

| Criterion | Label | Weight |
|---|---|---|
| Degree of Ecological Diversity | D | 25% |
| Adaptability Based on Local Conditions | A | 20% |
| Management of Spatial Sequences | S | 15% |
| Presentation of Visual Richness | R | 15% |
| Application of Landscaping Techniques | T | 15% |
| Minimization of Artificial Traces | N | 10% |

supplementary material. Every participant evaluates twelve distinct landscapes, with three scenes randomly selected for each approach. Twenty-one images are generated from viewpoints in Figure 6 for every landscape. Using a 5-point Likert Scale, participants score each criterion from 1 (lowest) to 5 (highest) for all landscapes. Finally, participants are asked to select the best one among all landscapes.

*5.3.2   Result and Analysis.* Thirty users participate in our study, comprising thirteen males and seventeen females with an average age of $\mu = 23.5$ and standard deviation $\sigma = 2.96$. Nine of them (30%) are familiar with landscape design. Ninety valid samples are collected for each approach, and the weighted scores are computed and presented in Figure 9a. The mean weighted score and standard deviations for the four approaches are 3.37(0.70), 3.39(0.69), 3.49(0.68), and 3.52(0.70), respectively. Though a one-way ANOVA test indicates no significant difference among the four groups of data ($p = 0.376$), Ours and Manual slightly outperform Random1 and Random2. It suggests that randomizing certain parts of our framework produces inferior results. Figure 9b shows the frequency of each approach being selected as the best. Despite our approach not being favored, it generates landscape scenes comparable to the manually crafted ones and exhibits satisfying plausibility.

In Figure 9c, we analyze scores for each criterion. Landscapes from Ours and Manual receive higher overall ratings. Manual scenes demonstrate significant advantages in criteria T and N, attributed to exquisite landscaping techniques and outstanding naturality. However, they receive lower scores for D, primarily due to fewer elements compared to the generated scenes, given the substantial effort required for manual operations. Compared to the crafted landscapes, our results excel in diversity and richness but lack aesthetics in other aspects. However, feedback reveals that some users fail to notice that most scenes are generated or cannot distinguish them from crafted ones, even after completing the study. Despite room for improvement, the generated landscapes can already compete with the manually designed ones in terms of quality.

## 6   Conclusion and Future Work

This paper introduces a framework leveraging the capabilities of LLMs and PCG for controllable landscape generation. Following a three-stage process, the framework can seamlessly convert any textual input into a batch of aesthetic landscapes tailored to user specifications. The performance analysis and ablation studies validate the plausibility of the proposed approaches, and the user study

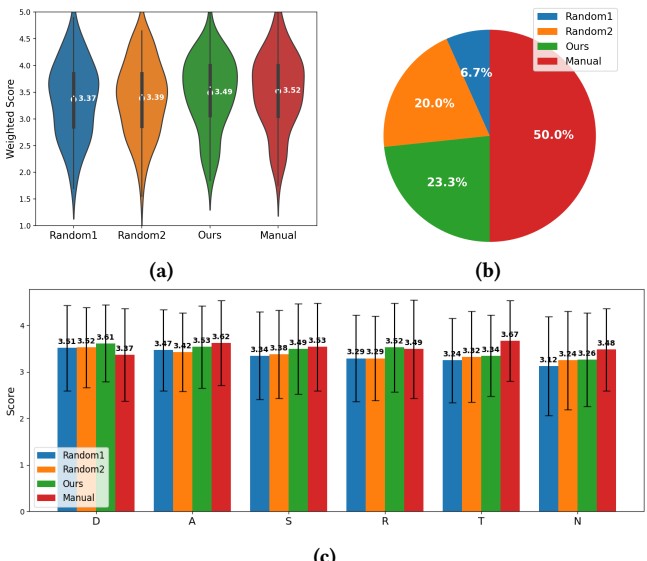

**(a)**                                      **(b)**

**(c)**

**Figure 9: In the user study, each criterion in Table 3 is rated from 1 to 5. (a) The weighted sum of scores considering all criteria. Landscapes by Ours and Manual receive better ratings. (b) The breakdown for selecting the best scene, indicating that the crafted scenes are the most favored. (c) Scores for each criterion. Our results excel in diversity and visual richness, whereas the designed scenes stand out in other criteria, especially techniques and naturality.**

demonstrates that the generated landscapes stand on par with those crafted by professional designers. The framework serves as a tool that enables efficient and controllable landscape generation for ordinary users, potentially enhancing design productivity. However, we still have the following limitations.

Firstly, the framework lacks dynamic effects and interactions in temporal and spatial dimensions. In real world, landscapes are vibrant components within a larger ecosystem. Landscapes are subject to dynamic influences and experience seasonal changes. Spatially, the landscape is intricately linked to its surroundings, encompassing natural factors like sunlight and artificial elements such as streets. We plan to contextualize the landscape within a broader framework and incorporate more dynamic interactions.

Secondly, the currently generated results are not yet comparable to well-designed real-world landscapes. For instance, the artificial traces of a grid cannot be removed by smoothing. Additionally, the limited rules, patterns, and element categories compromise flexibility in landscape arrangement. Our future direction involves introducing refined approaches and expanding the range of available assets to achieve more realistic results.

Finally, there remains room for enhancing controllability within the current framework. As discussed in Section 5.1.2, the conversion from user input to parameters occasionally suffers from inaccuracies. In the future, we will conduct more extensive experiments to understand the parameterization and enhance control.

## Acknowledgments

This work was supported by the National Key Research and Development Program of China (No. 2023YFF0905104), the National Natural Science Foundation of China (No. 62132012, 62361146854) and Tsinghua-Tencent Joint Laboratory for Internet Innovation Technology.

Shao-Kui Zhang is funded by Shuimu Tsinghua Scholar Program (No. 2023SM061), China Postdoctoral Science Foundation (No. 2024M751696), Postdoctoral Fellowship Program of CPSF (No. GZB20230353), Tsinghua University Student Research Training (No. 2421T0278, 2421T0277, 2411T0372, 2411T0371) and Young Elite Scientists Sponsorship Program by BAST (No. BYESS2024242).

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
