# OpenReview forum: "Controllable Procedural Generation of Landscapes"
_acmmm.org/ACMMM/2024/Conference — MM2024 Poster_

### Official Review · Reviewer_KAiZ · 2024-05-10

**Rating:** 6
**Confidence:** 3

**Summary:**

The manuscript presents a Procedural Content Generation (PCG) framework that simplifies the generation of tailored landscapes for ordinary users. The framework comprises three stages: parameterization, optimization of a layout plan, and generation of refined features.
In particular, it integrates LLM to input the description of the landscape from the user to control parameters that guide the generation process. It also leverages optimization and rule-based techniques and uses a grid to guide the landscape’s layout plan. Afterward, it mathematically smooths the transitions vertically and horizontally for a natural transition of the constituent landscape batches, i.e., terrains, vegetation, and roads, and arranges these elements to obtain an aesthetic and plausible landscape.

**Strengths:**

The paper is well-written, structured, and presented. The content includes cross-referencing, which enhances its coherence, facilitates understanding for readers, and makes it easy to follow
The authors conducted comprehensive experiments and evaluations
The authors concluded with insightful future works after identifying the limitations in their framework.

**Limitations:**

The visualization of Figure 2 can be improved for better understanding of the workflow in the framework

**Suitability:**

3

---

### Official Review · Reviewer_TG42 · 2024-05-21

**Rating:** 4
**Confidence:** 2

**Summary:**

This paper presents a framework that uses advanced language models to help users easily generate customized landscapes. By inputting simple text, users can create landscapes that match those designed by professional architects.

**Strengths:**

The paper is well-structured, featuring clear motivation and solution. The methodology that employs a language model to convert linguistic contexts into parameters is not novel, yet it appears to be effective.
The evaluation of the proposed work is comprehensive and robust, encompassing quantitative analyses, ablation studies, and user experiments.  The results depicted in the figures looks good.
The accompanying video effectively demonstrates the performance of the proposed method, and the supplementary documents provide detailed insights into the technical processes and evaluations.

**Limitations:**

However, I have several concerns regarding the technical content of the draft:

<1> Sections 4.2 and 4.3 lack detailed descriptions of the techniques used, despite references to further details in the supplementary material. The paper should contain all essential information necessary to replicate and evaluate the work.

<2> I recommend summarizing the variables required for landscape generation—such as terrain types, artificial elements, roads, and spots—at the beginning of the solution section (perhaps Section 3). This summary would help readers better understand the design parameters involved in creating a landscape.

<3> Key concepts, such as "plan" or "layout plan," should be formally defined to aid comprehension among readers who are not familiar with landscape design.

<4> The unity plug-in mentioned in Section 5.1 should be elaborated upon and integrated into the overall workflow of the proposed solution. I think it is a crucial component necessary for completing the landscape generation process.

**Suitability:**

3

---

### Official Review · Reviewer_nXZm · 2024-05-23

**Rating:** 5
**Confidence:** 3

**Summary:**

The paper proposes a controllable framework for the procedural generation of landscapes using Large Language Models (LLMs). The framework aims to enhance user accessibility and control by converting user text inputs into parameters that guide the generation process. The approach leverages optimization techniques and rule-based refinements to create landscapes that include terrains, roads, and various attributes. The framework is evaluated through extensive experiments and user studies, demonstrating its effectiveness in generating landscapes comparable to those crafted by experienced architects.

**Strengths:**

1. Innovative Integration of LLMs: The framework integrates state-of-the-art LLMs to convert user text inputs into generation parameters, significantly enhancing user accessibility and control over the procedural generation process.
2. Optimization Techniques: The use of optimization techniques, including a genetic algorithm for terrain and vegetation types and a heuristic algorithm for road generation, ensures the generation of aesthetically pleasing and functional landscapes.
3. Comprehensive Evaluation: The framework is thoroughly evaluated through performance analysis, ablation studies, and user studies, providing a robust validation of its effectiveness and comparability to manually crafted landscapes.
4. Practical Applications: The ability to generate diverse landscapes from simple text inputs has practical applications in various fields, including urban planning, gaming, and virtual environment design.

**Limitations:**

1. User Input Interpretation: The accuracy of converting user inputs into generation parameters can occasionally suffer, particularly when inputs are complex or negatively worded. This limitation impacts the overall controllability and precision of the generated landscapes.
2. Lack of Dynamic Effects: The current framework does not fully incorporate dynamic effects and interactions in temporal and spatial dimensions, limiting its realism in real-world scenarios where landscapes are subject to changes over time and interactions with surrounding elements.
3. Limited Rules and Patterns: The framework currently supports a limited set of rules, patterns, and element categories, which restricts the flexibility and diversity of landscape arrangements.

**Suitability:**

3

---

### Official Review · Reviewer_ijCm · 2024-06-03

**Rating:** 2
**Confidence:** 3

**Summary:**

This paper presents a controllable framework for generating controllable and aesthetic landscapes using large language models and procedural content generation. ​The framework allows users to describe their desired landscapes in plain text, which is then converted into parameters for landscape generation using a large language model.​ It leverages rule-based and optimization techniques (such as 2-D genetic algorithm) and refinements (such as mathematical smoothing) to generate a general plan for terrains, roads, and various attributes. ​ The plan is refined through smoothing and arranging elements for aesthetic landscapes. ​ The framework supports batch generation of diverse landscapes and has been shown to produce landscapes comparable to those crafted by experienced architects validated by a user study. ​ The paper also discusses the limitations of the current framework as lacking dynamic interactions and suggests future improvements for higher controllability.

**Strengths:**

Following are the strengths of the paper:

1. Clarity and Thoroughness: The paper is easy-to-read and provides a clear and detailed explanation of the proposed framework for generating landscapes. ​ The methodology is well-described making it easy for readers to understand the framework.

2. Practical Application and Relevance: The paper addresses a practical problem in landscape design by proposing a controllable framework for procedurally generating landscapes. By integrating language models and optimization techniques, the framework offers a solution that enhances productivity and allows ordinary users to generate plausible landscapes tailored to their specifications. ​ The relevance of the research to the field of landscape design and the potential impact on the industry make this paper valuable.

**Limitations:**

Following are the limitations of the paper:
1. Novelty: While the proposed framework for generating landscapes is well-explained and the integration of language models and procedural content generation is interesting, I have some concerns regarding the novelty of the approach. ​ It appears that the framework mostly combines GPT4 and existing techniques such as genetic and heuristic algorithms, mathematical smoothing with the exception of a novel 'evolution' method adapted in the genetic algorithm. ​For ex:- It is not clear as to how this framework is different or better compared to 3D-GPT.  It would be helpful if you could further highlight the novel aspects or improvements over existing methods to clearly establish the novelty of your framework.

2. Insufficient evaluation: Secondly, although you mention extensive experiments in the paper, I believe there is room for further evaluation to strengthen the validity of your findings.
  a. While the performance analysis and ablation studies provide valuable insights, it would be beneficial to include additional quantitative metrics or comparisons with other state-of-the-art methods to demonstrate the superiority of your framework (For ex:- 3D GPT). ​
  b. If the novelty lies in the framework, there needs to be quantitative evaluation with state-of-the-art. Also applicable If the novelty lies in the adapted genetic algorithm.
  c. To evaluate the LLM component, its ability to interpret user input is evaluated in terms of accuracy in determining the existence of a lake as per user input.
  d. Additionally, the user study comparing the landscapes generated by your framework with those crafted by experienced architects is intriguing, but it would be valuable to provide more details on the methodology and results of this study. ​


Overall, I appreciate the clarity of your paper and the potential of your framework in addressing the challenges of landscape generation. ​ However, I believe further emphasis on the novelty of the approach and a more thorough evaluation would significantly strengthen the contribution of your work.

**Suitability:**

3

---

### Meta-Review · Area_Chair_QzAa · 2024-07-03

**Recommendation:** Accept (Poster)
**Confidence:** 4

**Metareview:**

This paper received three positive ratings and one negative rating as the final ratings. The major concerns of the negative comment are the novelty and insufficient evaluation. Regarding novelty, the reviewer asked for clarifications on the novelty in comparison to 3D-GPT, for which the authors provided reasonable and convincing responses. The AC agrees that these two works address different tasks and have different focuses. Regarding limited evaluations, the AC thinks that comprehensive evaluation remains a common challenge for generation problems, and the evaluation presented in the paper is sufficient to justify its effectiveness. Nevertheless, the authors should improve the clarity of the evaluation methods in the final version. Overall, the AC agrees with the majority and recommends acceptance.